# Shikimate Kinase Plays Important Roles in Anthocyanin Synthesis in Petunia

**DOI:** 10.3390/ijms232415964

**Published:** 2022-12-15

**Authors:** Junwei Yuan, Shiwei Zhong, Yu Long, Jingling Guo, Yixun Yu, Juanxu Liu

**Affiliations:** Guangdong Key Laboratory for Innovative Development and Utilization of Forest Plant Germplasm, College of Forestry and Landscape Architecture, South China Agricultural University, Guangzhou 510642, China

**Keywords:** SK, shikimate pathway, anthocyanin, metabolome, petunia

## Abstract

In plants, the shikimate pathway is responsible for the production of aromatic amino acids L-tryptophan, L-phenylalanine, and L-tyrosine. L-Phenylalanine is the upstream substrate of flavonoid and anthocyanin synthesis. Shikimate kinase (SK) catalyzes the phosphorylation of the C3 hydroxyl group of shikimate to produce 3-phosphate shikimate (S3P), the fifth step of the shikimate pathway. However, whether SK participates in flavonoid and anthocyanin synthesis is unknown. This study characterized the single-copy *PhSK* gene in the petunia (*Petunia hybrida*) genome. *PhSK* was localized in chloroplasts. *PhSK* showed a high transcription level in corollas, especially in the coloring stage of flower buds. Suppression of *PhSK* changed flower color and shape, reduced the content of anthocyanins, and changed the flavonoid metabolome profile in petunia. Surprisingly, *PhSK* silencing caused a reduction in the shikimate, a substrate of *PhSK*. Further qPCR analysis showed that *PhSK* silencing resulted in a reduction in the mRNA level of *PhDHQ/SDH*, which encodes the protein catalyzing the third and fourth steps of the shikimate pathway, showing a feedback regulation mechanism of gene expression in the shikimate pathway.

## 1. Introduction

In plants, the shikimate pathway provides carbon skeletons for the aromatic amino acids (AAAs) L-tryptophan, l-phenylalanine, and Ltyrosine, which can be turned into an array of aromatic secondary metabolites, such as flavonoids, alkaloids, and lignins, and is a bridge between carbohydrate metabolism and protein metabolism [1,2]. It uses erythrose-4-phosphate (E4P) and phosphoenolpyruvate (PEP), the intermediate products of the pentose phosphate pathway and glycolysis pathway, to synthesize chorismate under the catalysis of six key enzymes through seven steps of catalytic reaction [2,3,4]. Genes involved in this pathway have been identified and isolated in microorganisms, fungi, and plants. The six key enzymes are as follows: 3-deoxy-D-arabino-heptulosonate 7-phosphate synthase (DAHPS), 3-dehydroquinate synthase (DHQS), 3-dehydroquinate dehydratase (DHQ)–shikimate dehydrogenase (SDH), shikimate kinase (SK), 5-enolpyruvylshikimate 3-phosphate synthase (EPSPS), and chorismate synthase (CS) (Appendix A) [5].

In plants, several studies have reported the function of some genes of the shikimate pathway. Injury, pathogen invasion, and environmental stress could increase the expression of *DAHPS* in plants, and its overexpression could enhance the resistance of plants to the environment [6,7,8,9]. In *Arabidopsis thaliana*, *DHQS* expression was not well correlated to phenylpropanoid production [10]. Silencing of *NtDHQ/SHD-1* caused growth inhibition and a reduction in AAAs in tobacco [11]. EPSPS, as the target enzyme of herbicides, has been well studied in Arabidopsis [12], *Lolium perenne* [13], and tobacco [14]. The first cloned plant *CS* gene came from *Euglena gracilis* (Schaller et al., 1991), and petunia (*Petunia hybrida*) *PhCS* silencing resulted in growth inhibition, flower deformity, and whitening [15].

In *Escherichia coli*, SK has two isozymes, AroL and AroK. AroL is common in prokaryotes [16,17,18,19]. However, in yeast and fungi, SK is a domain of the multifunctional enzyme AROM, which catalyzes DAHP to generate EPSP directly [20,21]. Among higher plants, the *LeSK* gene of tomato (*Lycopersicon esculentum*) was isolated first in 1992. The nitrogen end of the LeSK protein contained a chloroplast transport peptide. Its amino acid sequence is very similar to that of the bacterial SK protein, and LeSK had catalytic activity in the process of plant tissue culture. Synthesized LeSK is transported to chloroplast and processed to be an active mature protein [22]. The rice (*Oryza sativa*) genome contains three *SK* genes, *OsSK1*, *OsSK2*, and *OsSK3*, and three OsSKs were all localized in the chloroplast [23]. Arabidopsis genome had two *SK* genes, *AtSK1* and *AtSK2* [24]. The same as ATP-driven enzymes, the activity of SK isolated from spinach (*Spinacia oleracea*) was regulated by the energy level of cells [25]. SK is necessary for the biosynthesis of EPSP with shikimate as the substrate because it is the only known enzyme that is capable of phosphorylating the C3 hydroxyl group of shikimate to produce 3-phosphate shikimate (S3P) [8].

Anthocyanins are flavonoid pigments and play important roles in reproduction and protection against various abiotic and biotic stresses [26]. Anthocyanins are primarily generated in the cytoplasm and endoplasmic reticulum before being transferred to vacuoles for storage and accumulation [27,28]. Phenylalanine is the upstream substrate of anthocyanin synthesis (Appendix A). Under the catalysis of three key enzymes, phenylalanine is converted to 4-coumarinyl-CoA, the precursor of anthocyanin synthesis [29]. However, whether SK participates in flavonoid and anthocyanin synthesis is unknown.

In this study, a single-copy *PhSK* was characterized in the petunia genome, and *PhSK* was localized in chloroplasts. Among the examined organs, *PhSK* had the highest expression in corollas. Suppression of *PhSK* changed flower color and shape, reduced the content of anthocyanins, and changed the flavonoid metabolome profile in petunia. In addition, *PhSK* silencing caused a reduction in the shikimate, a substrate of *PhSK*.

## 2. Results

### 2.1. Isolation and Sequence Analyses of PhSK

We used cDNA sequences of *AtSK1* (accession no. NM_201778.3) and *AtSK2* (accession no. NM_202986.2) from Arabidopsis as the query in BLAST and searched the *Petunia axillaris* draft genome sequence v1.6.2 (https://solgenomics.net/organism/Petunia_axillaris/genome, accessed on 1 November 2022), and only one petunia *SK*, *PhSK*, was recovered. We further searched several previously published petunia transcriptomes [30,31,32], and still, only one *PhSK* was recovered. The same full-length *PhSK* cDNA sequence was obtained from petunia ‘Ultra’. *PhSK* encodes a putative 283-amino-acid protein with a predicted molecular weight of 31.9 kDa.

Eight mRNA sequences of *SK*s from five plant species, petunia *PhSK* (Peaxi162Scf00359g00118.1), *Solanum lycopersicum SlSK* (Solyc04g051860.3.1, https://solgenomics.net/organism/Solanum_lycopersicoides/genome, accessed on 1 November 2022), Arabidopsis *AtSK1* and *AtSK2*, *Oryza sativa OsSK1* (NM_001401978.1), *OsSK2* (XP_015641676.1) and *OsSK3* (XP_015636368.1), and *Vitis vinifera VvSK* (NM_001281087.1), were obtained. The CDS size of the eight *SK*s was similar, ranging from 852 bp to 927 bp. As shown in Appendix A, the size of nuclear gene sequences of the eight *SK*s ranged from 2178 bp to 7371 bp, among which *AtSK2* was the smallest and *SlSK* is the largest. The number of introns in eight *SK*s varied from 7 to 9 (Appendix A).

Multiple sequence alignments of the *SK*s of petunia, *Nicotiana tabacum*, *S. lycopersicum*, Arabidopsis, *O. sati*va, and *E. coli* are shown in Appendix A. The SK amino acid sequences in the plant showed high similarity. The deduced amino acid sequence of *PhSK* had 85.7%, 89.0%, 67.3%, 64.1%, 57.6%, 55.9%, 64.7%, 30.7%, and 32.7% identities with NtSK (NP_001312965.1), SlSK (NP_001234112.1), AtSK1 (NP_001077936.1), AtSK2 (NP_195664.2), OsSK1 (XP_015626759.1), OsSK2 (XP_015641676.1), OsSK3 (XP_015636368.1), AroL (WP_000193393.1), and AroK (WP_000818618.1), respectively. In addition, the N-terminal and C-terminal regions of these *SK*s showed low similarity, while the middle region showed high similarity.

As a member of the nucleoside monophosphate kinase (NMP) family, *PhSK* is characterized by its Walker A-motif, Walker B-motif, LID domain, and shikimate-binding domain, as shown in Appendix A. The 111th to 118th amino acids of *PhSK* are Walker A-motif, which has the conserved sequence G-X-X-G-X-G-K-T/S for forming the P-ring to bind the β-Phosphate group of nucleotides. The 178th to 183rd amino acids of *PhSK* are Walker B-motif, which has the consensus sequences V-X-A/S-T-G-G for binding nucleotides. The 214th to 229th amino acids of *PhSK* are the LID domain of SK-binding ATP. The 135th to 164th amino acids of *PhSK* are the shikimate-binding domain, which binds shikimate [33,34,35].

To elucidate the evolutionary relationship among SK proteins in plants, a phylogenetic tree was constructed from SK amino acid sequences of *E. coli* and 12 species of plants using the TBtools software. The phylogenetic analysis showed that plant *SK*s belong to a small family, generally including one to three members (Appendix A).

### 2.2. PhSK Protein Localization in Chloroplasts

In order to determine the subcellular localization of *PhSK* in plant cells, green fluorescent protein (GFP) was fused to the full-length C-terminal of *PhSK* to create a pSAT-*PhSK*-GFP vector, which was then transferred to petunia leaf protoplasts. Fluorescent signals were only detected in chloroplasts by confocal microscopy after incubation for 16–24 h. The results showed that *PhSK* protein was localized in chloroplasts (Figure 1).

### 2.3. PhSK Expression

The expression of *PhSK* in different plant organs and in different stages of flower or leaf development was examined by quantitative RT-PCR (qPCR). The transcription level of *PhSK* was the highest in corollas and the lowest in stems. During leaf growth, the expression of *PhSK* first increased and then decreased. When the flower bud length was 2 cm, *PhSK* expression reached the peak, then decreased, and then increased again during flower development (Figure 2).

### 2.4. Phenotype of PhSK-Silenced Petunia Plants

To study the function of *PhSK*, the pTRV2-*PhSK* vector was constructed for *PhSK* silencing. The pTRV2-GFP vector containing a 717-bp fragment of the *GFP* served as the control. A total of 25 to 30 petunia seedlings were used for infection.

One month later, pTRV2-*PhSK*-treated plants showed shorter stem internode lengths compared with the control plants (Figure 3A–D), while pTRV2-*PhSK*-treated leaves did not show visible change.

*PhSK* silencing resulted in shorter pedicels and sepals, and the diameter of the corolla tubes in *PhSK*-silenced plants was significantly shortened compared with the control, while their length remained unchanged (Table 1; Figure 4A,B). The diameter of the corollas of *PhSK*-silenced plants was 85.6% of that of the control. The color of *PhSK*-silenced corollas was lighter compared with the control. In *PhSK*-silenced plants, the adaxial plane of corollas exhibited light speckles, and the abaxial plane of the corollas was light in color. The number of light hairs on the abaxial plane of the corollas of *PhSK*-silenced plants increased, making the abaxial plane of the corollas have a velvety texture and luster (Figure 4A,B). The anthocyanin content of *PhSK*-silenced corollas was significantly reduced compared with the control (Figure 5A).

pTRV2-*PhSK* treatment significantly reduced the *PhSK* mRNA level in corolla, which was only 66.7% of the control (Appendix A). We further examined SK activities and found that *PhSK* silencing dramatically decreased the activities of SK in corollas compared with the control (Figure 5B).

### 2.5. Changes in the Corolla Flavonoid Metabolome Profile Induced by PhSK Silencing

To further analyze the effects of *PhSK* silencing on the content of the flavonoid metabolites, a widely targeted metabolomics analysis of corollas was performed with UPLC and tandem MS. To ensure that the samples were taken from *PhSK*-silenced corollas, the lighter portion of the corolla was collected and verified by qPCR assays. The metabolites in the samples were analyzed quantitatively and qualitatively by MS based on the KEGG database (https://www.genome.jp/kegg/pathway.html, accessed on 1 November 2022), MWDB database (Metware Biotechnology Co., Wuhan, China), and MRM. A total of 298 flavonoid metabolites and 5 tannins were identified (Appendix A).

In order to screen the differential metabolites, the criteria used for the screening included a fold change value ≥ 2 or ≤0.5 and a VIP value ≥ 1. In this study, 102 metabolites were significantly changed with a high level of repeatability (Figure 6A, Appendix A). A KEGG database was used to categorize all differential metabolites. The differential metabolites in *PhSK*-silenced corollas were mainly enriched in isoflavonoid biosynthesis, flavonoid biosynthesis, flavone and flavonol biosynthesis, biosynthesis of secondary metabolites, and anthocyanins biosynthesis (Figure 6B; Appendix A).

A total of 99 flavonoid metabolites changed significantly, of which 22 were upregulated and 77 were downregulated (Figure 6A; Appendix A). The top ten downregulated metabolites were Rhamnetin, Chrysoeriol-7-O-(6″-acetyl)glucoside, Kaempferol-3-O-arabinoside, Oroxin A, Genistein-7-O-galactoside-rhamnose, Epigallocatechin, Eupatilin-7-O-glucoside, Kaempferol, Quercetin-3-O-(6″-p-Coumaroyl)glucoside, and Quercetin-3-O-(6″-p-Coumaroyl)galactoside. The top ten upregulated metabolites were Limocitrin-3-O-galactoside, Petunidin-3-O-(6″-O-feruloyl)rutinoside-5-O-glucoside, Quercetin-3-O-(6″-acetyl)glucosyl-(1→3)-Galactoside, Isorhamnetin-7-O-glucoside, Rhamnetin-3-O-Glucoside, 6-Methoxykaempferol-3-O-glucoside, Isorhamnetin-3-O-Glucoside, Pinocembrin, Tricin-7-O-(6″-O-malonyl)glucoside, and Isorhamnetin-3-O-(6″-acetylglucoside) (Appendix A).

In addition, 44 anthocyanin metabolites were detected. A total of 6 anthocyanin metabolites were downregulated, and only one, Petunidin-3-O-(6″-O-feruloyl)rutinoside-5-O-glucoside, was upregulated with a fold change of 2.02 (Table 2). The 6 downregulated anthocyanins included Cyanidin-3-O-(2′′-O-glucosyl)rutinoside, Cyanidin-3-O-(6″-O-malonyl)sophoroside-5-O-glucoside, Cyanidin-3,3′-di-O-glucoside-7-O-(6″-O-caffeoyl)glucoside, Cyanidin-3-O-glucoside, Cyanidin-3-O-sophoroside-5-O-glucoside, Delphinidin-3-O-rutinoside-7-O-glucoside, and Petunidin-3-O-(6″-O-feruloyl)rutinoside-5-O-glucoside (Table 2, Appendix A).

In addition, two-thirds (22/33) of flavones (Appendix A), all three differential isoflavones (Appendix A), most (34/41) flavonols (Appendix A), two-thirds (2/3) of flavanonols (Appendix A), and three-quarters (6/8) of flavanones (Appendix A) were significantly downregulated in *PhSK*-silenced plants compared with the control. These results showed that *PhSK* silencing could reduce the total content of flavonoids.

### 2.6. PhSK Silencing Reduces the Shikimate Content in Corollas

Shikimate serves as a substrate of *PhSK*, and we measured the content of shikimate in corollas. As shown in Figure 5C, unexpectedly, the content of shikimate in *PhSK*-silenced corollas was significantly decreased compared with the control.

### 2.7. Effect of PhSK Silencing on the Expression of Some Structural Genes of the Shikimate Pathway and Anthocyanin Synthesis Pathway

The effects of *PhSK* silencing on the transcript levels of *PhDHQ/SDH* (Peaxi162Scf01067g00113.1), *PhEPSPS1* (Peaxi162Scf00959g00022.1), *PhCHSA* (Peaxi162Scf00047g01225.1), and *PhF3′5′H* (Peaxi162Scf00150g00218.1) in corollas were analyzed by qPCR. The results showed that *PhSK* silencing significantly increased the mRNA level of *PhCHSA* and significantly decreased the mRNA levels of *PhDHQ/SDH*, *PhEPSPS1,* and *PhF3′5′H* (Figure 7).

## 3. Discussion

Generally, *SK* is a small gene family in many plants, and there are two and three *SK* members in Arabidopsis and rice genomes, respectively. In this study, there was only one *PhSK* member in the petunia genome. *PhSK* exhibited 67.3% and 57.6% identity with AtSK1 and OsSK1, respectively, indicating that the amino acid sequences of SK are conserved in higher plants. In addition, there are four conservative domains in the middle of *PhSK*, and these domains are the most important functional domains of SK [33,34,35].

It has been demonstrated that the AAA synthesis pathway mostly occurs in plastids, but some of the intermediates in this pathway (e.g., shikimate and chorismate) are exported to the cytosol and are used as precursors for the synthesis of proteins and other compounds (e.g., phenylpropanoids, indole compounds, and alkaloids) [8]. In tobacco, the DHQ/SDH family includes two members, NtDHQ/SDH1 and NtDHQ/SDH2. The NtDHQ/SDH1-YFP signal was confined to the plastids, whereas NtDHQ/SDH-2 was localized in the cytosol [11]. A chloroplast import assay and the presence of the cTP sequence (ChloroP1.1) indicated that petunia PhEPSPS1 and PhCM1 were plastid-localized proteins [36], while PhCM2 was not localized in chloroplasts [37]. Petunia PhCS was localized in peroxisomes and chloroplasts [15]. *S. lycopersicum* prephenate aminotransferase (SlPAT) was localized in chloroplasts [38]. Three petunia ADTs were localized in plastids [39]. Shikimate was exported from the plastids and conjugated with p-coumaroyl-coenzyme A (CoA) with the function of hydroxycinnamoyl-CoA shikimate hydroxycinnamoyl transferase in the cytosol, thereby generating p-coumaroyl shikimate [40]. In this study, SK was exclusively localized in plastids, so the phosphorylation of shikimate only occurs in the plastid in petunia.

*SK* is an upstream gene of the AAA synthesis pathway in plants, and AAAs play an important role in plant growth and development [8]. Antisense RNA-mediated *DAHPS* silencing blocked shikimate biosynthesis in the plastids of potato cells and resulted in delayed growth, reduced stem length and width, and reduced stem lignin content in potato plants [41]. RNAi suppression of *NtDHQ/SDH* delayed plant growth in tobacco [11]. In our previous study, suppression of petunia PhCS, catalyzing the last step of the shikimate pathway, led to a dwarf phenotype, small flower, and yellow deformed leaves [15]. These studies indicate that a general restriction of the shikimate pathway could block the development of vegetative organs. However, in this study, *PhSK* silencing only caused a shortened length of the internodes in petunia, showing only a slight effect on the growth and development of vegetative organs. The reason for the shortened internode length and small flower may be that *PhSK* silencing led to the decrease in the content of lignin, the downstream metabolite of the AAA synthesis pathway [8]. Lignin is essential for the maintenance of structural integrity, stem elongation, and the formation of leaves and flowers [42]. *PhSK* silencing did not result in severe phenotypic changes in vegetative organs as *PhCS* silencing did [15], and it is possible that *PhCS* plays a more important role in vegetative organ development than *PhSK*. PhCS is localized in chloroplasts and in peroxisomes [15], which is different from *PhSK*. In addition, it cannot be ruled out that *PhSK* silencing may lead to changes in secondary metabolites of vegetative organs.

In this study, *PhSK* silencing significantly reduced the content of anthocyanins and changed the flavonoid metabolome profile, indicating that *PhSK* plays an important role in the synthesis of flavonoids, including anthocyanins. Similarly, petunia *PhCS* silencing resulted in abnormal flower development and a reduction in the total anthocyanin content [15].

In this study, *PhSK* silencing resulted in significant downregulation of the expression of its upstream gene *PhDHQ/SDH* and its downstream gene *PhEPSPS1*, indicating a feedback regulation mechanism of gene expression in the shikimate pathway. The downregulation of *PhDHQ/SDH* expression could explain the reason for the decrease in shikimate in *PhSK*-silenced plants. In addition, the mRNA level of *PhCHSA* was significantly upregulated, indicating that *PhCHSA* mRNA level may be regulated by the feedback of anthocyanin synthesis. Similarly, *PhCS* silencing slightly increased the expression of *PhEPSPS1*, *PhCM1*, and *PhCHSJ*, indicating the existence of feedback regulation of the expression of these genes by anthocyanins or other products of the shikimate pathway [15]. In previous AAA synthesis pathway studies, *N. silvestris CM1* was activated by tryptophan but inhibited by phenylalanine and tyrosine in a feedback mechanism [43]. In addition, the feedback-resistant forms of anthranilate synthase have been reported in potato, *N. otophora*, *N. tabacum,* and Arabidopsis [44,45].

## 4. Materials and Methods

The Sanli Horticultural Company of Guangzhou provided the petunia ‘Ultra’ seeds, and the seedlings were cultivated in a greenhouse (23 ± 2 °C, 60% relative humidity, and 14 h light/10 h dark cycle) [46]. The leaves, stems, and roots were harvested when the plants were about 25 cm tall in the vegetative stage. The flowers were collected at anthesis (corollas 90° reflexed) and were placed in water immediately. Each 0.2 g sample was wrapped with foil, immediately placed in liquid nitrogen, and kept at −80 °C until used [47]. Unless otherwise stated, three biological replicates from independent collection and extraction of tissues were used in all studies.

### 4.1. RNA Extraction, RT-PCR, and Cloning of the Petunia PhSK Gene

According to the manufacturer’s instructions, total RNA was extracted from the roots, stems, leaves, and corollas with an R4151B-HiPure Plant RNA Kit B (R4151B, Magen, China). Following the instructions, petunia mRNA was reverse-transcribed with an HiScript III 1st Strand cDNA Synthesis Kit (+gDNA wiper) (R312-01/02, Vazyme, China). With specific primers (Appendix A) based on the petunia genome’s sequences (https://solgenomics.net/organism/Petunia_axillaris/genome, accessed on 1 November 2022), full-length *PhSK* (Peaxi162Scf00359g00118.1) cDNAs were isolated.

### 4.2. Sequence Analysis

Multiple sequence alignments were created with the DNAMAN (version 5.2.2, Lynnon Biosoft, San Ramon, CA, USA) software, and a phylogenetic tree was created with TBtools (version 1.098765) software [48]. The BLAST network server of the National Center for Biotechnology Information (NCBI) (https://www.ncbi.nlm.nih.gov/, accessed on 1 November 2022) was used to identify nucleotides and translated amino acids. With the accession from related studies and the results of BLAST on NCBI and Solgenomics, different *SK* nuclear gene and coding sequence (CDS) sequences from different species were obtained. The nuclear gene sequences were used for analyzing the gene structure of *SK*s on the GSDS2.0 website (http://gsds.gao-lab.org/index.php, accessed on 1 November 2022).

### 4.3. Subcellular Localization

Subcellular localization analysis was carried out according to the previously reported protocol [49]. The *GFP* gene-containing pSAT-1403TZ vector (https://www.ncbi.nlm.nih.gov/nucleotide/56553541, accessed on 1 November 2022) was utilized to create the *PhSK*-GFP construct. (Tzfira et al. 2005). A full-length PCR amplification of the *PhSK* CDS sequence was performed, and the PCR products were cloned into the pSAT-1403TZ vector, which utilizes the CaMV 35S promoter to drive GFP fusions. The results of each recombinant vector were identified by PCR and confirmed by sequencing analysis. The sequences of the primers are described in Appendix A.

As previously mentioned, ethylene glycol was used to separate and prepare petunia leaf protoplasts [50]. After being incubated in the dark for 24 h, the protoplasts were visualized by the Zeiss (http://www.zeiss.com, accessed on 1 November 2022) LSM710 microscope. Excitation/emission wavelengths for GFP and chlorophyll were 488/535 nm and 488/637 nm, respectively.

### 4.4. Quantitative Real-Time PCR Assays

Quantitative real-time PCR (qPCR) was performed according to the previous methods [51]. The primers used for qPCR are shown in Appendix A. According to the instructions, 1 μL of cDNA, 10 μL of Taq Pro Universal SYBR qPCR Master Mix (Q712, Vazyme, China), 0.4 μL of forward primer, 0.4 μL of reverse primer, and 8.2 μL of sterile water were mixed to prepare each 20 μL reaction. The samples were subjected to the following thermal cycling procedures: a DNA predenaturation stage lasting 30 s at 95 °C, an amplification stage taking 40 cycles of 10 s at 95 °C and 30 s at 60 °C, and the final stage lasting 15 s at 95 °C, 60 s at 60 °C, and 15 s at 95 °C to create a melting curve. The assays employed 3 different cDNAs from the same time point that were generated from 3 different RNAs, and each study was performed in triplicate. To confirm their identities, the amplicons underwent electrophoresis analysis and sequencing. The quantification was constructed using Pfaffl’s threshold cycle (Ct) value analysis [51]. It was executed following the Minimum Information for Publication of Quantitative Real-Time PCR Experiments guidelines when conducting the analyses (Bustin et al., 2009; Tan et al., 2014). For the quantification of cDNA abundance, *cyclophilin* (*CYP*) (no. EST883944) was chosen as the internal reference gene [52]. The data in the study are a representation of the relative expression values determined by *CYP*. Appendix A provides information on the sequences of each primer used in the qPCR analysis. Each treatment’s three biological duplicates were analyzed.

### 4.5. Agroinoculation of pTRV2 Vectors

To create the pTRV2-*PhSK* vector, specific primers (Appendix A) were used for the amplification of the 254-bp sequence of the 3′ regions of *PhSK* by PCR. The pTRV2-GFP vector containing a 717-bp fragment of the *GFP* was previously constructed as the control [15]. We performed BLAST searches of the *Petunia axillaris* draft genome sequence v1.6.2 using the inserted *GFP* sequence as the query, and no gene or fragment had homology with *GFP*. As previously mentioned, pTRV1 and pTRV2-GFP or pTRV2-*PhSK* vectors were transferred to the *Agrobacterium tumefaciens* GV3101 strain [47,53,54]. Cultivated in the liquid YEP medium [55] with 50 mg L^−1^ of kanamycin and 200 μM of acetosyringone, the *A. tumefaciens* cells were grown at 28 °C for 8–10 h. Then, *A. tumefaciens* cells were collected and resuspended to an OD_600_ of 2–3 in the inoculation buffer containing 200 M acetosyringone, 10 mM MES, and 10 mM MgCl_2_ (pH 5.5). *A. tumefaciens* carrying pTRV1 was diluted 1:1 with *A. tumefaciens* carrying pTRV2-GFP or pTRV-*PhSK* after 1 h of incubation at 28 °C. After the apical meristems were removed, the wound surface of 4-week-old petunia plants was subsequently treated with roughly 300 μL of this mixture. Each vector was used to inoculate 25 to 30 plants.

### 4.6. Assays of SK Activity

A total of 0.1 g of fresh corolla used for the SK activity assay was collected. The fresh corolla was ground with 9 mL of PBS basic (pH 7.2–7.4 and 0.01 mol L^−1^). After centrifuging the mixture at 4000 rpm for 30 min, the supernatant was stored for the following analysis. The activity of SK was analyzed with the Shikimate Kinase Activity Kit (Jingkang, Shanghai, China). Standard curves were established using the following standard concentrations: 0 U L^−1^, 7.5 U L^−1^, 15 U L^−1^, 30 U L^−1^, 60 U L^−1^, and 120 U L^−1^. Prepared samples were put into an ELISA plate containing an HRP-labeled reagent after being diluted five times. The plate was then kept at 37 °C for 60 min. The dish was carefully washed with detergent to ensure that there was no liquid residue. After adding a chromogenic agent to the ELISA plate, it was kept in the dark for 15 min. After adding the termination solution, Varioskan LUX (Thermo Scientific, Waltham, MA, USA) was used to spectrophotometrically measure the absorbance of the extracts at 450 nm. Three biological replicates were analyzed for each treatment. The precise instructions are provided by Shikimate Kinase Kit (Jingkang, Shanghai, China).

### 4.7. Anthocyanin Extraction and Measurement

A total of 0.2 g of petunia corollas was used to extract anthocyanins as previously reported [56,57]. Petunia corollas were ground into a powder with liquid nitrogen. The powder was transferred into a 50 mL centrifuge tube with 30 mL acidic methanol containing 1% HCl (*v*/*v*). The mixture was kept at 4 °C for 2 h in the dark. After centrifuging at 10,500 rpm for 10 min, the supernatant was used for measuring the absorption values. The absorption values of the extract were measured at A_530_ and A_657_ to calculate the anthocyanin concentration using the formula A_530_ − 0.25A_657_, which eliminates the influence of chlorophyll. Three biological replicates were analyzed for each treatment.

### 4.8. Widely Targeted Metabolomics Analysis

The corollas of petunia were harvested, freeze-dried, and powdered. A total of 0.1 g of powder was soaked overnight at 4 °C in 1.0 mL of 70% aqueous methanol. After centrifuging the extract at 10,000× *g* for 10 min, the supernatant was filtered through a 0.22 m pore size microporous membrane for liquid chromatography–tandem mass spectrometry (LC-MS/MS) analysis.

The metabolites were analyzed using ultra-performance liquid chromatography (UPLC) (Shim-pack UFLC SHIMADZU CBM30A, http://www.shimadzu.com.cn/, accessed on 1 November 2022) and MS/MS (AB SCIEX 6500 QTRAP) under the following conditions outlined by Li and Song [58]: column, water ACQUITY UPLC HSS T3 C18 1.8 μm, 2.1 mm × 100 mm; mobile phase, the aqueous phase was ultrapure water (0.04% acetic acid), and the organic phase was acetonitrile (0.04% acetic acid); water/acetonitrile gradient, 95:5 *V*/*V* for 0 min, 5:95 *V*/*V* for 11.0 min, 5:95 *V*/*V* for 12.0 min, 95:5 *V*/*V* for 12.1 min, and 95:5 *V*/*V* for 15.0 min; flow rate, 0.4 mL/min; column temperature, 40 °C; and injection volume, 2 μL. The electrospray ionization (ESI) temperature was 500 °C, the MS voltage was 5500 V, the curtain gas (CUR) was 25 psi, and the collision-induced dissociation (CAD) parameter was set to high. Each ion pair was scanned for detection in triple quadrupole mode (QQQ) using the optimal decompression potential (DP) and collision energy (CE) [59].

### 4.9. Qualitative and Quantitative Determination of Metabolites

The metabolites of the samples were analyzed qualitatively and quantitatively by MS utilizing the self-built MetWare database (MWDB) (MetWare Company, Wuhan, China) (http://www.metware.cn/, accessed on 1 November 2022) and multiple reaction monitoring (MRM) [59]. The isotope and repeated signals were eliminated during the qualitative analysis of the material using secondary spectral data. Utilizing MRM with QQQ MS, the metabolites were quantified. Before screening the precursor ions of the target substance, the ions matching compounds with various molecular weights were removed. Additionally, the precursor ions were broken down into fragment ions in the collision cell, and then the characteristic fragment ions were chosen using QQQ filtering. The quantitative results are more precise and repeatable as a result of these processes. The resulting mass spectrum peaks of the metabolites underwent peak area integration; moreover, the mass spectral peaks of the metabolites from different samples were integrated [59].

### 4.10. Shikimate Measurement

Samples were prepared according to the methods of the SK activity assays above. The content of shikimate was measured using the Plant Shikimate Kit (Coibo Bio, Shanghai, China). Standard curves were established using the following standard concentrations: 0 ng/mL, 2.5 ng mL^−1^, 5 ng mL^−1^, 10 ng mL^−1^, 20 ng mL^−1^, and 40 ng mL^−1^. Prepared samples were put into an ELISA plate with an HRP-labeled reagent after being diluted five times. The plate was then kept at 37 °C for 60 min. The detergent cleansed the dish carefully to make sure that no liquid was left. After adding a chromogenic agent to the ELISA plate, it was kept in the dark for 15 min. After adding the termination solution, Varioskan LUX (Thermo Scientific, Waltham, MA, USA) was used to spectrophotometrically measure the absorbance of the extracts at 450 nm. Three biological replicates were analyzed for each treatment. The precise instructions are provided by Shikimate Kit (Jingkang, Shanghai, China).

### 4.11. Statistical Analyses

One-way analysis of variance (ANOVA) and Duncan’s multiple range test (DMRT) with at least three replicates were used to statistically analyze the data. *p* values under 0.05 were regarded as significant.

## 5. Conclusions

There is only one *PhSK* member in the petunia genome, and *PhSK* is localized in chloroplasts. This study provides genetic evidence that *PhSK* plays an important role in the synthesis of flavonoid and anthocyanin metabolites.

## Figures and Tables

**Figure 1 ijms-23-15964-f001:**
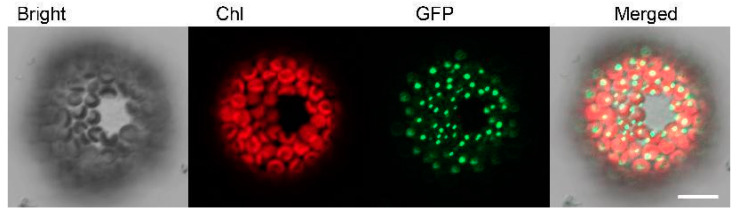
Subcellular location of *PhSK*. *PhSK* C-terminal GFP fusion proteins were transiently expressed in petunia protoplasts and visualized by confocal microscopy. Chl, chlorophyll. Scale bars: 5 μm. Images were processed by Zen 2010 (version 6.0, Carl Zeiss Microscopy GmbH, Germany) software.

**Figure 2 ijms-23-15964-f002:**
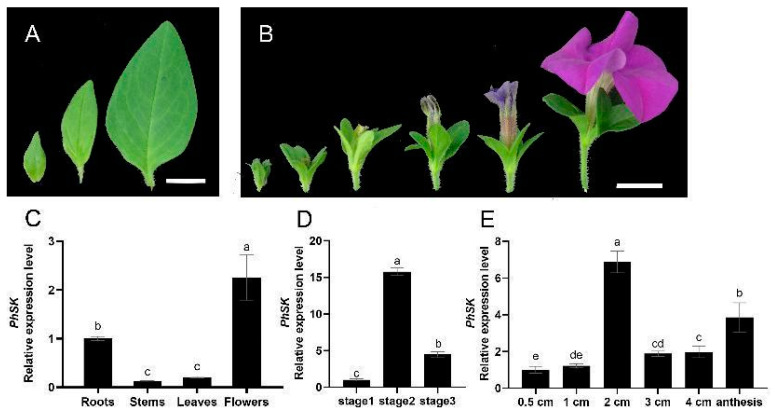
Expression patterns of *PhSK* determined by quantitative real-time PCR. (**A**) Three developmental stages of petunia leaves: stage 1 (young leaves, 1.0 cm), stage 2 (growth leaves, 2.5 cm), and stage 3 (mature leaves, 4.0 cm). (**B**) Six stages of petunia flower buds: 0.5 cm, 1 cm, 2 cm, 3 cm, 4 cm, and anthesis. (**C**–**E**) Expression of *PhSK* in leaves, corollas, roots, and stems (**C**), in leaves in three stages (**D**), and in corollas during flower development (**E**). (**A**) Bar = 1 cm, (**B**) bar = 2 cm. Letters a–e mean significant differences between data sets. The datas with same letter mean non-significance. The datas with different letters mean significant differences.

**Figure 3 ijms-23-15964-f003:**
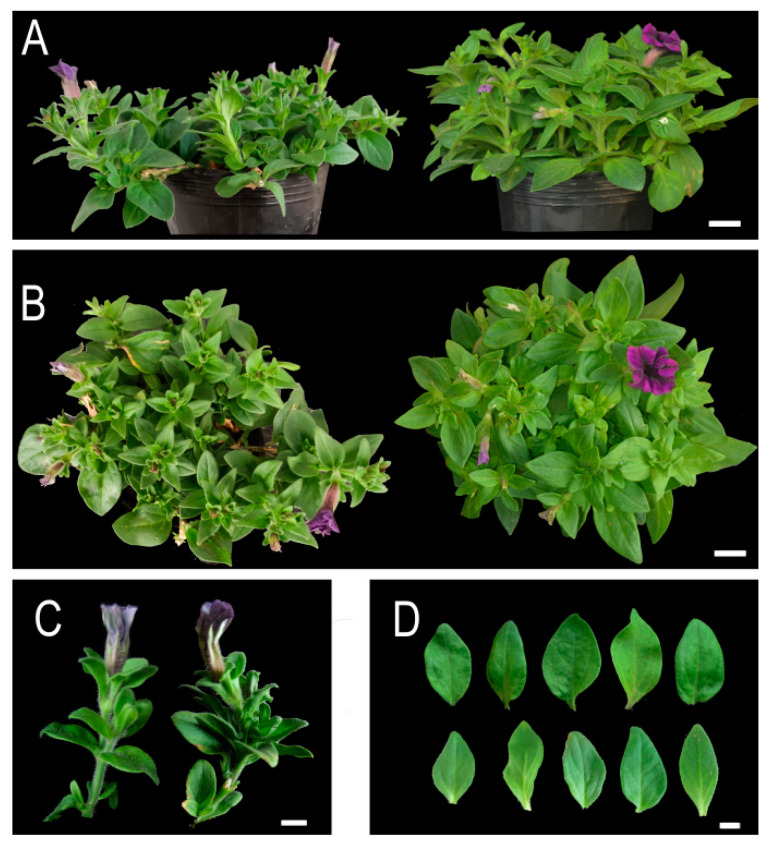
Plant phenotype of *PhSK*-silenced petunia vegetative organs. (**A**) Four-week-old control plant (**left**) and *PhSK*-silenced plant (**right**). (**B**) The vertical view of the 4-week-old control plant (**left**) and *PhSK*-silenced plant (**right**). (**C**) Internodes of the control plant (**left**) and *PhSK*-silenced plant (**right**). (**D**) Leaves of the control plant (**top**) and *PhSK*-silenced plant (**bottom**). (**A**,**B**) Bar = 2 cm, (**C**,**D**) bar = 1 cm.

**Figure 4 ijms-23-15964-f004:**
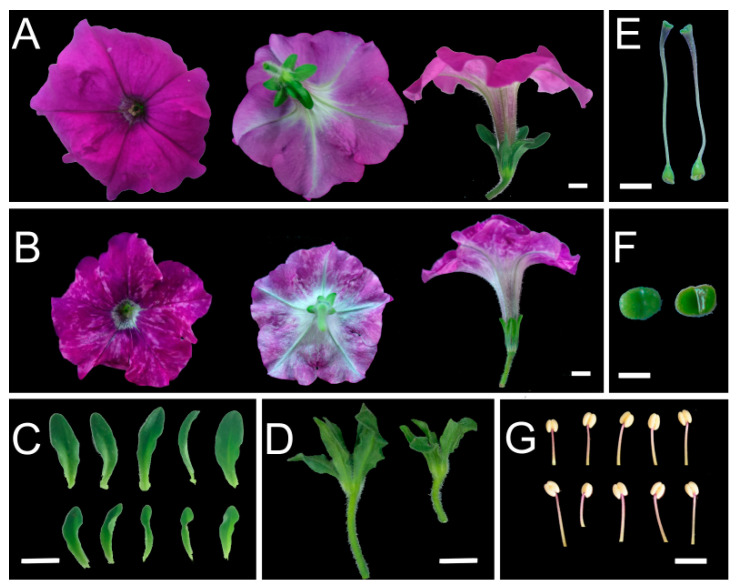
Effect of *PhSK* silencing on flowers. (**A**,**B**) Top view, back view, and side view of control flower (**A**) and *PhSK*-silenced flower (**B**). (**C**) Calyces of the control plants (**top**) and *PhSK*-silenced plants (**bottom**). (**D**) Sepals of the control plants (**left**) and *PhSK*-silenced plants (**right**). (**E**) Pistils of the control plants (**left**) and *PhSK*-silenced plants (**right**). (**F**) Stigmas of the control plants (**left**) and *PhSK*-silenced plants (**right**). (**G**) Stamens of the control plants (**top**) and *PhSK*-silenced plants (**bottom**). (**A**–**D**) Bar = 1 cm, (**E**) bar = 0.5 cm, (**F**,**G**) bar = 0.2 cm.

**Figure 5 ijms-23-15964-f005:**
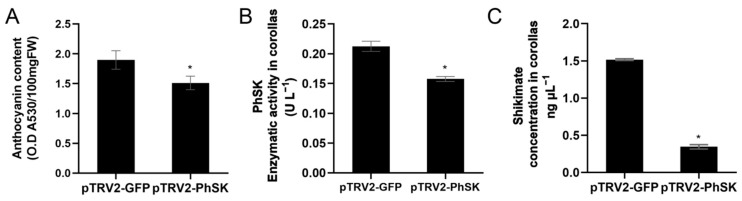
Effects of pTRV2-*PhSK* treatment on the anthocyanin content (**A**), enzymatic activity of SK (**B**), and shikimate content (**C**). The data are presented as the means ± SDs (*n* = 3). The statistical analysis was performed using the one-way analysis of variance (ANOVA) followed by Duncan’s multiple range test (DMRT) with three biological replicates. * indicates significant differences at the *p* ≤ 0.05 level.

**Figure 6 ijms-23-15964-f006:**
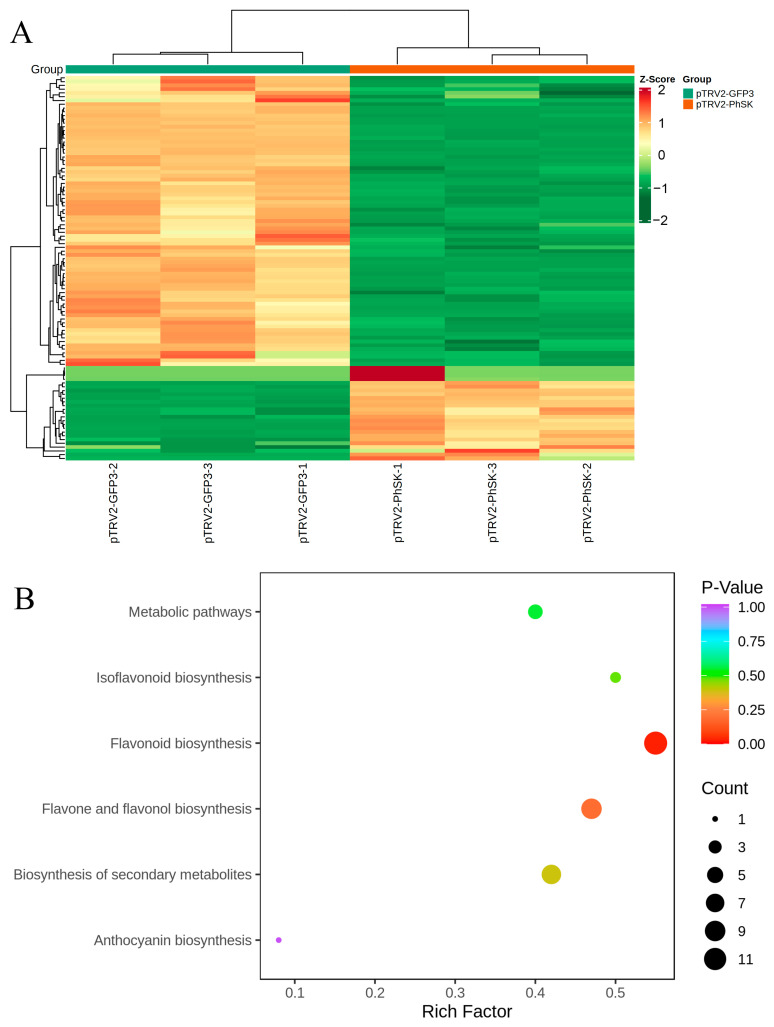
Changes in the corolla flavonoid metabolome profile induced by *PhSK* silencing. (**A**) Heat map of the differential metabolites in *PhSK*-silenced compared with control petunia corollas. Green indicates a decrease in differentially expressed metabolites, and red indicates an increase in differentially expressed metabolites. (**B**) KEGG enrichment analysis of the differentially abundant metabolites in *PhSK*-silenced and control petunia corollas. Data were processed by ggplot2 (version 3.3.0, https://CRAN.R-project.org/package=ggplot2, accessed on 1 November 2022) software.

**Figure 7 ijms-23-15964-f007:**
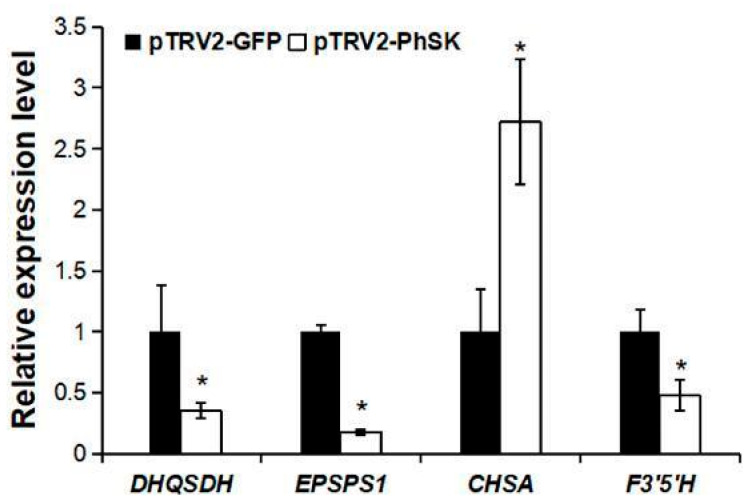
Effects of pTRV2-*PhSK* treatment on the expression of *PhDHQSDH*, *PhEPSPS1*, *PhCHSA*, and *PhF3′5′H* in corollas. *Cyclophilin* (*CYP*, accession no. EST883944) was used as the internal reference gene for the quantification of cDNA abundance. The data are presented as the means ± SDs (n = 3). The statistical analysis was performed using the one-way analysis of variance (ANOVA) followed by Duncan’s multiple range test (DMRT) with three biological replicates. * indicates significant differences at the *p* ≤ 0.05 level.

**Table 1 ijms-23-15964-t001:** Effects of *PhSK* silencing on petunia plant growth.

	pTRV2-GFP	pTRV2-*PhSK*	pTRV2-*PhSK*/ pTRV2-GFP (%)
Length of pedicels (cm)	2.71 ± 0.67	1.58 ± 0.47 *	58.30
Length of sepals (cm)	2.05 ± 0.20	1.69 ± 0.20 *	82.43
Width of sepals (cm)	0.58 ± 0.09	0.56 ± 0.16	96.55
Length–width ratio of sepals (cm)	26.57 ± 2.88	35.11 ± 4.49 *	132.14
Circumference of corolla tube (cm)	2.24 ± 0.19	1.85 ± 0.25 *	82.59
Length of corolla tube (cm)	3.12 ± 0.09	3.23 ± 0.27	103.53
Diameter of corollas (cm)	6.73 ± 0.37	5.76 ± 0.59 *	85.59
Length of internodes (cm)	2.61 ± 0.55	1.52 ± 0.37 *	58.24

Data are the means ± SEs from 15 to 20 samples. Statistical analysis was performed using Student’s *t*-test with 15 to 20 replicates. * indicates significant differences at the *p* ≤ 0.05 level.

**Table 2 ijms-23-15964-t002:** Seven anthocyanins changed significantly in *PhSK*-silenced and control petunia corollas.

Compounds	VIP	Fold Change	Type
Cyanidin-3-O-(2″-O-glucosyl)rutinoside	1.09	0.46	down
Cyanidin-3-O-(6″-O-malonyl)sophoroside-5-O-glucoside	1.04	0.47	down
Cyanidin-3,3′-di-O-glucoside-7-O-(6″-O-caffeoyl)glucoside	1.14	0.48	down
Cyanidin-3-O-glucoside (Kuromanin)	1.09	0.48	down
Cyanidin-3-O-sophoroside-5-O-glucoside	1.12	0.48	down
Delphinidin-3-O-rutinoside-7-O-glucoside	1.14	0.49	down
Petunidin-3-O-(6″-O-feruloyl)rutinoside-5-O-glucoside	1.09	2.02	up

The criteria used for this screening included a fold change value ≥ 2 or ≤0.5 and a VIP value ≥ 1.

## Data Availability

Not applicable.

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
