# Peer review of "Shikimate Kinase Plays Important Roles in Anthocyanin Synthesis in Petunia"

_ijms, 2022, doi:10.3390/ijms232415964_

Round 1
Reviewer 1 Report
The introduction should be rewritten and focused on the current study. Currently it starts out with discussion of aromatic amino acids which is hardly relevant to the study.
Figure 1B , the anthocyanin biosysnthetic pathways is unnecessary detail. Anthycyanins are one of many metabolites downstream of SK.
The primers for the PhSK gene cloning must be shared in the methods. Similarly, the primers used for RT-PCR are essential to share in the methods.
Figure 8 is missing, which is critical to interpreting the paper.
How were the metabolites accurately quantified without standards? I am skeptical and there isn't detail in the MS to convince me otherwise.
I am left with questions about the silencing and its impact. on pathway genes. Part of this is from a lack of empty vector control, missing figure 8 and questions of about whether the insert had any sequence similarities to the other impacted genes, etc.
This manuscript is in need of language editing as well as basic attention to spellcheck.
Author Response
Reviewer 1
1.The introduction should be rewritten and focused on the current study. Currently it starts out with discussion of aromatic amino acids which is hardly relevant to the study.
Response: We have carefully checked the logic of the introduction and made modifications. At the beginning of the introduction, we started from the shikimic acid pathway, the key point of the study, and modified the part about aromatic amino acids. (lines 37–43) Thank you.
2.Figure 1B , the anthocyanin biosysnthetic pathways is unnecessary detail. Anthycyanins are one of many metabolites downstream of SK.
Response: We have revised Figure 1 as follows and we have placed it in supplemental data (Figure S1) according to the advices of reviewer 2. Thank you.
3.The primers for the PhSK gene cloning must be shared in the methods. Similarly, the primers used for RT-PCR are essential to share in the methods.
Response: All primers used in this study have been mentioned in the methods, and the primer sequences are provided in Supplementary Table 1-3.
4.Figure 8 is missing, which is critical to interpreting the paper.
Response: The missing Figure 8 has been added to the updated manuscript. Because the original Figure 1 has been placed in the supplemental data (Figure S1), Figure 8 has now become Figure 7.
5.How were the metabolites accurately quantified without standards? I am skeptical and there isn't detail in the MS to convince me otherwise.
Response: Widely targeted flavanoid metabolomics analysis was performed by MetWare Ltd. Co. (Wuhan, China). We added the Acknowledgements section, to thank MetWare Co., Ltd. (Wuhan, China) for the metabolome services. Widely targeted flavanoid metabolomics analysis is based on the self-built MetWare database (MWDB) (MetWare Company, Wuhan, China) (http://www.metware.cn/) and multiple reaction monitoring (MRM), using five parameters of substance detection to detect the relative content of metabolites in different samples and obtain qualitative and quantitative data of substances (Chen et al, 2013). It is relative quantification, that is, relative quantification of the mass spectrum peak area. After obtaining the mass spectrum analysis data of metabolites in different samples, we integrate the peak area of all substances' mass spectrum peaks, and integrate and correct the mass spectrum peaks of the same metabolite in different samples. The peak area of each chromatographic peak represents the relative content of corresponding substances (Chen et al., 2013). Widely targeted flavanoid metabolomics analysis has been applied in a large number of publications (Sang et al., 2022; Zhao et al., 2021; Zhao et al., 2020; Zhong et al., 2020; and so on). The methods of Widely targeted flavanoid metabolomics analysis in this study were provided in lines 365-390
6.I am left with questions about the silencing and its impact on pathway genes. Part of this is from a lack of empty vector control, missing figure 8 and questions of about whether the insert had any sequence similarities to the other impacted genes, etc.
Response: The added Figure 7 and Figure S5 in the updated version provided the results of qPCR assay, which showed that the relative expression level of PhSK was significantly reduced and that PhSK silencing changed the expression of the pathway genes.
More than three biological repeated experiments have yielded the consistent results of impact on pathway genes. Similiarly, PhCS silencing slightly increased the expression of PhEPSPS1, PhCM1, and PhCHSJ, indicating the existence of feedback regulation of the expression of these genes by anthocyanins or other products of the shikimate pathway [Zhong et al, 2020]. In previous AAA synthesis pathway studies, N. silvestris CM1 was activated by tryptophan, but inhibited by phenylalanine and tyrosine in a feedback mechanism (Goers and Jensen 1984). In addition, the feedback-resistant forms of anthranilate synthase have been reported in potato, N. otophora, N. tabacum and Arabidopsis (Schmid and Amrhein 1995; Swinney et al, 1997). (Lines 271-276)
We performed BLAST searches of the Petunia axillaris draft genome sequence v1.6.2 using the inserted GFP sequence as the query, and no gene or fragament has homology with GFP. (L333-335) pTRV2-GFP treated plants did not show significant difference compared with the uninfected control. In addition, previous study discussed the application of VIGS technology in petunia in detail, and suggested pTRV2-GFP as a control instead of pTRV2 empty vector (Broderick and Jones, 2014).
7.This manuscript is in need of language editing as well as basic attention to spellcheck.
Response: We carefully checked the whole manuscript, tried our best to revise the manuscript and performed language editing.
Reference:
Broderick, S.R.; Jones, M.L. An Optimized Protocol to Increase Virus-Induced Gene Silencing Efficiency and Minimize Viral Symptoms in Petunia. Plant Mol. Biol. Rep. 2014, 219-233. doi:10.1007/s11105-013-0647-3.
Chen, W.; Gong, L.; Guo, Z.; Wang, W.; Zhang, H.; Liu, X.; Yu, S.; Xiong, L.; Luo, J. A Novel Integrated Method for Large-Scale Detection, Identification, and Quantification of Widely Targeted Metabolites:Application in the Study of Rice Metabolomics. Mol. Plant. 2013, 6, 1769-1780. doi:10.1093/mp/sst080.
Goers, S.K.; Jensen, R.A. The differential allosteric regulation of two chorismate-mutase isoenzymes of Nicotiana silvestris. Planta 1984, 162, 117-124. doi:10.1007/BF00410207.
Jones, J.D.; Henstrand, J.M.; Handa, A.K.; Herrmann, K.M.; Weller, S.C. Impaired wound induction of 3-deoxy-D-arabino-heptulosonate-7-phosphate (DAHP) synthase and altered stem development in transgenic potato plants expressing a DAHP synthase antisense construct.. Plant Physiol. 1995, 108, 1413-1421. doi:10.1104/pp.108.4.1413.
Sang, L.; Chen, G.; Cao, J.; Liu, J.; Yu, Y. PhRHMs play important roles in leaf and flower development and anthocyanin synthesis in petunia. Physiol. Plant. 2022, 174, e13773. doi:10.1111/ppl.13773.
Schmid, J.; Amrhein, N. Molecular organization of the shikimate pathway in higher plants. Phytochemistry 1995, 39, 737-749. doi:10.1016/0031-9422(94)00962-S.
Swinney, D.C.; Mak, A.Y.; Barnett, J.; Ramesha, C.S. Differential allosteric regulation of prostaglandin H synthase 1 and 2 by arachidonic acid. The Journal of biological chemistry 1997, 272, 12393-12398. doi:10.1074/jbc.272.19.12393.
Zhao, H.; Chen, G.; Sang, L.; Deng, Y.; Gao, L.; Yu, Y.; Liu, J. Mitochondrial citrate synthase plays important roles in anthocyanin synthesis in petunia. Plant science : an international journal of experimental plant biology 2021, 305, 110835. doi:10.1016/j.plantsci.2021.110835.
Zhao, H.; Zhong, S.; Sang, L.; Zhang, X.; Chen, Z.; Wei, Q..; Chen, G.; Liu, J.; Yu, Y.; Melzer, R. PaACL silencing accelerates flower senescence and changes the proteome to maintain metabolic homeostasis in Petunia hybrida. J. Exp. Bot. 2020, 71, 4858-4876. doi:10.1093/jxb/eraa208.
Zhong, S.; Chen, Z.; Han, J.; Zhao, H.; Liu, J.; Yu, Y. Suppression of chorismate synthase, which is localized in chloroplasts and peroxisomes, results in abnormal flower development and anthocyanin reduction in petunia. Sci. Rep. 2020, 10, 10846. doi:10.1038/s41598-020-67671-6.

Reviewer 2 Report
This manuscript describes the functions of Shikimate kinase (SK) in regulating anthocyanin accumulation in petunia. The authors used the cDNA sequences of Arabidopsis AtSK1 and AtSK2 to perform homologous searching for possible homologs in Petunia, leading to the identification of the only SK in Petunia, named PhSK. Further, they showed that PhSK was localized in chloroplasts and its transcripts were abundant in flowers and low in stems and leaves. To further elucidate the functions of PhSK in Petunia, the authors utilized the VIGS approach to reduce the expression levels of PhSK in Petunia. The pTRV2-PhSK treated plants showed shorter stem internodes, shorter floral organs, including pedicels, sepals, corolla tubes, and corollas, and a reduction of anthocyanin content. Eventually, they determined changes in the corolla flavonoid metabolome profile in pTRV2-PhSK treated plants. In addition, pTRV2-PhSK treated plants also reduced the shikimate content, implying a negative feedback regulation in the shikimate pathway. This study links SK functions with anthocyanin content and flavonoid-related products. However, some concerns are described in the following.
1. Figure 1 should be in the supplemental data instead of the official figure.
2. It is better to validate the degree of the VIGS in this study.
3. Explain why pTRV2-PhSK treated plants result in a decrease of stem internodes and a reduction of floral organs.
4. There are some English errors in the text. The authors should revise them.
Author Response
1.Figure 1 should be in the supplemental data instead of the official figure.
Response: We have placed it the supplemental data (Figure S1). Thank you.
2.It is better to validate the degree of the VIGS in this study.
Response: The added Figure S5 in the updated version provided the results of qPCR assay, which showed that the relative expression level of PhSK was significantly reduced.
3.Explain why pTRV2-PhSK treated plants result in a decrease of stem internodes and a reduction of floral organs.
Response: Antisense RNA-mediated DAHPS silencing blocked shikimate biosynthesis in the plastids of potato cells and resulted in delayed growth, reduced stem length and width, and reduced stem lignin content in potato plants (Jones et al., 1998). RNAi suppression of NtDHQ/SDH delayed plant growth in tobacco (Li et al., 2007). In our previous study, suppression of petunia PhCS, catalyzing the last step of the shikimate pathway, led to a dwarf phenotype, small flower, and yellow deformed leaves (Zhong et al., 2020). These studies indicates that a general restriction of shikimate pathway could block the development of vegetative organs. The reason for the shortened internode length and small flower may be that PhSK silencing led to the decrease of the content of lignin, the downstream metabolites of AAA synthesis pathway (Maeda et al., 2012). Lignin is essential for the maintenance of structural integrity, stem elongation, and the formation of leaves and flowers (Lewis et al., 1990). (lines 246-257).
4.There are some English errors in the text. The authors should revise them.
Response: We carefully checked the whole manuscript, tried our best to revise the manuscript and performed language editing.
Reference:
Lewis, N.G.; Yamamoto, E. Lignin: occurrence, biogenesis and biodegradation. Annu. Rev. Plant Physiol. Plant Mol. Biol. 1990, 41. 455-496. doi:10.1146/annurev.pp.41.060190.002323.
Li, D.; Hofius, D.; Hajirezaei, M.; Fernie, A.R.; Börnke, F.; Sonnewald, U. Functional analysis of the essential bifunctional tobacco enzyme 3-dehydroquinate dehydratase/shikimate dehydrogenase in transgenic tobacco plants. J. Exp. Bot. 2007, 58, 2053-2067. doi:10.1093/jxb/erm059.
Maeda, H.; Dudareva, N. The shikimate pathway and aromatic amino Acid biosynthesis in plants. Annu. Rev. Plant Biol. 2012, 63, 73-105. doi:10.1146/annurev-arplant-042811-105439.
Zhong, S.; Chen, Z.; Han, J.; Zhao, H.; Liu, J.; Yu, Y. Suppression of chorismate synthase, which is localized in chloroplasts and peroxisomes, results in abnormal flower development and anthocyanin reduction in petunia. Sci. Rep. 2020, 10, 10846. doi:10.1038/s41598-020-67671-6.
